# Oxidative Stress and High-Mobility Group Box 1 Assay in Dogs with Gastrointestinal Parasites

**DOI:** 10.3390/antiox11091679

**Published:** 2022-08-28

**Authors:** Michela Pugliese, Ettore Napoli, Salvatore Monti, Vito Biondi, Elena Zema, Annamaria Passantino

**Affiliations:** 1Department of Veterinary Sciences, University of Messina, Via Giovanni Palatucci, 98168 Messina, Italy; 2Veterinary Practitioner, Via Caserta Crocevia, 25, 89124 Reggio Calabria, Italy

**Keywords:** gastrointestinal nematodes, reactive oxidative metabolites, antioxidant barrier, thiol groups of plasma compounds, high-mobility group box 1, dogs

## Abstract

This study aimed to evaluate the concentration of reactive oxidative metabolites, the antioxidant barrier, thiol groups of plasma compounds, and high-mobility group box 1 in shelter dogs naturally infected with helminths. In addition, the correlation between clinical signs and oxidative stress was investigated. Sixty-six (41 male and 25 female) adult mixed-breed dogs housed in a shelter with the diagnosis of gastrointestinal nematodes (i.e., *Ancylostoma* spp., *Uncinaria stenocephala*, *Toxocara canis*, *Toxascaris leonina*, or *Trichuris vulpis*) were enrolled in Group 1 (G1) and twenty healthy adult dogs were included in Group 2 (G2), which served as the control. A clinical assessment was performed using a physician-based scoring system. Oxidative stress variables and high-mobility group box 1 were assessed and compared by the means of unpaired *t*-tests (*p* < 0.05). Spearman’s rank correlation was performed to calculate the correlation between oxidative stress variables, high-mobility group box 1, hematological parameters, and clinical signs. The results showed statistically significant values for reactive oxidative metabolites, thiol groups of plasma compounds, and high-mobility group box 1 in G1. Negative correlations between thiol groups and the number of red cells and hemoglobin were recorded. These preliminary results support the potential role of oxidative stress and HGMB-1 in the pathogenesis of gastrointestinal helminthiasis in dogs.

## 1. Introduction

Dogs are continuously exposed to parasitic infections, with gastrointestinal helminths being particularly prevalent [1,2,3,4,5,6,7,8,9] and with some of them being of zoonotic concern [10]. A recent epidemiological survey conducted throughout Europe has shown that intestinal nematode infections remain a common occurrence in dogs. This has also been demonstrated by Rehbein and collaborators [11] in a study conducted on 1390 owned dogs, which reported that more than a third of them were shedding eggs/cysts of endoparasites.

The risk of parasitic infection is higher in animals living in communities (e.g., kennels or shelters) and/or with access to the outdoors compared to owned dogs [12]. However, some owned dogs, those used for work in particular (for instance, hunting dogs), can also be threatened by a wide range of parasite species. In a survey conducted in southern Italy, a frequency of intestinal parasites was observed that ranged from 14.6% to 48.8%, and the frequency was considerably higher in hunting dogs living in kennels, with 84% of mixed infestations [13].

In general, whipworms and hookworms are the species with the highest frequencies observed in Europe [11,13]. Hookworms are among the most prevalent canine helminths worldwide. Dog feces that are deposited in crowded cities can contaminate the environment with parasite eggs and larvae, thereby contributing to zoonotic transmission; in fact, dogs may play a central role in the spread of some helminths with a zoonotic potential [11,14].

Parasites can have consequences on host fitness, which depend on the animal’s immune system competence and the animal’s age. In fact, gastrointestinal helminths in young animals can cause severe intestinal disorder and gastroenteritis; conversely, infections that occur in adult dogs can occur without clinical signs due to age-associated immunity [15]. However, despite the clinical presentation of the disease, helminth infections in dogs may cause pathophysiological changes involving inflammation, oxidative stress, and changes to the protein, lipid, and iron metabolism and pancreatic function, which are all related to the activation of the immune system [16]. In fact, to defend the host from infection, the immune system may release toxic agents that can generate a state of oxidative stress if they are not adequately neutralized by the antioxidant system [16].

Oxidative stress occurs when there is an imbalance between the production of oxidant substances and the antioxidant defenses of the organism [17]. The continuous and excessive production of oxidizing agents leads to the consumption of antioxidant defenses.

High-mobility group box 1 (HMGB-1) is a nuclear protein that is involved as a fundamental co-factor in transcriptional regulation, and that is secreted into the extracellular environment following an apposite stimulus. Extracellular HMGB-1 acts as an alarmin that contributes to the intensification of inflammatory responses, thus interrelating with endothelial cells and triggering the release of pro-inflammatory mediators [18,19,20,21,22,23,24]. Moreover, the central role of HMGB-1 in response to oxidative stress in humans has been documented in several pathological conditions [25].

The role of oxidative stress in dogs has been studied in patients affected by *Leishmania infantum* and *Ehrlichia canis* [26,27]. However, to the authors’ best knowledge, no study has evaluated oxidative stress induced by helminths in dogs; therefore, the aim of this study was to evaluate the concentration of reactive oxidative metabolites (R-OOHs), the antioxidant barrier (OXY), thiol groups of plasma compounds (SHp), and HGMB-1 in shelter dogs naturally infected with helminths. An additional aim was to investigate the correlation between clinical signs and oxidative stress.

## 2. Materials and Methods

### 2.1. Ethical Statement

Animals were handled and sampled following institutional approval from the Ethics Committee of the Department of Veterinary Sciences, University of Messina (approval number: 49, 21 March 2021).

The study was carried out in accordance with the recommendations of the European Council Directive, 2010/63/EU, on the protection of animals used for scientific purposes and the Italian legislation (D.lgs. 26/2014, L. n. 281/1991, and L.R. 15/2000).

The included dogs were enrolled from shelters during routine sanitary visits. Each dog shelter’s administration was informed of the research purpose and clinical procedures involving the dogs, and written informed consent was signed before sample collection as proposed in the national guidelines for animal welfare.

### 2.2. Animals

Mixed-breed dogs housed in three different rescue shelters located in Sicily, southern Italy were screened for the presence of endoparasites. Each animal underwent a clinical visit (please see below) and individual fecal samples were collected and analyzed through a flotation technique using a solution of sodium nitrate and sugar (1.300 N/m3 SPG) for the detection of protozoan oocysts and helminth eggs [28]. The morphological identification of the eggs was performed at the genus/species level using morphological keys [29,30,31].

Animals were included in the study only if the presence of gastrointestinal helminths was detected within 3 days before the beginning of the study, in which case they were enrolled in Group 1 (G1). Moreover, 20 healthy dogs that tested negative for endoparasites were included in the study and served as the control group (G2). Dogs having received a pharmacological treatment against gastrointestinal parasites within six months before the start of the study were excluded.

### 2.3. Clinical Score (CS)

A clinical assessment, including the determination of body weight, was performed using a physician-based scoring system. Eleven variables were considered. The scores for each variable were added to obtain a maximum of eleven points (Table 1). The determination of body weight was measured on fasted animals, in the morning, by using a digital scale.

### 2.4. Blood Sample Collection and Analysis

Whole blood samples were collected at 09.00 a.m. from a peripheral vein (jugular or cephalic) from fasting dogs using a standard technique, and the samples were stored in a 2.5 mL anticoagulant (K_3_ EDTA) tube and 2.5 mL cloth activator tube.

Sera samples obtained after centrifugation at 3000× *g* × 20 min were stored at −80 °C until the analysis.

An investigation of oxidative stress variables (R-OOHs, OXY, and SHp) was performed with an ultraviolet spectrophotometer (Slim SeaC, SeaC, Florence, Italy). For the detection of radicals, the spin-trapping method, which examines the production of stable radicals derived from the reaction of oxygen metabolites using chemical acceptor molecules (commonly called “spin-trapping agents”), was used. D-ROMs were assayed to estimate the biomolecules produced by the peroxidation of lipids, amino acids, proteins, and nucleic acids during tissue damage [23,26].

The obtained values were correlated to the reaction intensity with chromogen peroxides and were expressed in Carratelli units (1 CARR U = 0.08 mg% hydrogen molecules photometrically detected).

OXY assessed the capability of the serum barrier to counteract the oxidative action induced by an excess of hypochlorous acid solution, with the plasma within a water solution and the concentration of residual unreacted radicals detected after the oxidant action. The intensity of the colored complex was inversely correlated with the total antioxidant capacity. A surplus of hypochlorous acid radicals present after the oxidative reaction indicated a reduction in the plasma barrier; conversely, a reduction in the values was directly correlated with an injury of the plasma barrier due to oxidation [24,27].

SHp was estimated as the capability of thiol groups to produce a colored complex in the presence of 5,5-dithiobis-2-nitrobenzoic acid. Low SHp concentrations were suggestive of an altered competence of the thiol antioxidant barrier.

HGMB-1 was assayed with EDTA plasma using a commercially available human ELISA kit (IBL-International, Hamburg, Germany) previously validated for canine species [32].

Hematological measurements and calculations, including the red blood cell count (RBC), packed-cell volume (PCV), hemoglobin concentration (Hgb), mean corpuscular hemoglobin (MCH), mean corpuscular hemoglobin concentration (MCHC), white blood cell count (WBC), and blood platelet count (PLT), were carried out on K3EDTA samples.

Clinical chemistry profile analyses were performed on serum samples obtained by centrifugation to assay the blood urea nitrogen (BUN), creatinine (CREA), total proteins (TP), alanine-amino transferase (ALT), aspartate-amino transferase (AST), albumin (ALB), and globulin (GLOB). All samples were analyzed in duplicate by the same operator.

### 2.5. Statistical Analysis

Statistical analyses were performed using the SPSS for Windows package, version 22.0.

The Kolmogorov–Smirnov test was used to assess the normal data distribution. Descriptive statistics were determined and expressed as means (±standard deviations [SD]) for normally distributed variables.

An unpaired *t*-test was applied to evaluate the differences between G1 and G2. The correlation between the considered variables was determined using Spearman’s rank test. A *p*-value lower than 0.05 was considered statistically significant.

## 3. Results

The G1 group was composed of 66 (41 male and 25 female) adult mixed-breed dogs with a mean age of 1.2 ± 2.5 y.o. and a body weight ranging between 7 and 42 Kg.

Throughout the copromicroscopic analysis, the presence of several gastrointestinal nematodes was identified, including *Ancylostoma* spp., *Uncinaria stenocephala*, *Toxocara canis*, *Toxascaris leonina*, and *Trichuris vulpis*.

The G2 group was composed of 20 (eight male and twelve female) adult mixed-breed dogs with a mean age of 4.5 ± 3.4 y.o. and a body weight ranging between 11 and 28 Kg.

All dogs belonging to G1 presented with several clinical signs, including pale mucous membranes (48/66; 72.7), ill thrift (39/66; 59.1%), a poor haircoat (35/66; 53%), abdominal swelling (29/66; 43.9%), lethargy (19/66; 12.5%), diarrhea (18/66; 27%), vomiting (15/66; 22.7%), dehydration (8/66; 5.3%), hyperthermia (6/66; 10%), and melena (6/66; 10%). The mean clinical score was 4.9 ± 3.1.

The laboratory findings showed the presence of anemia (18/66; 27%), leukocytosis (37/66; 56%), and hypoalbuminemia (23/66; 34.8) (Table 2).

When comparing the two groups, statistically significant differences in the RBC (*p* = 0.042), Hgb (*p* = 0.038), PCV (*p* = 0.016), WBC (*p* = 0.027), ALB (*p* = 0.045), and TP (*p* = 0.046) were observed (Table 2).

The dogs in G1 showed statistically higher values of R-OOHs (*p* = 0.014), SHp (*p* = 0.047), and HGMB-1 (*p* = 0.046) than those in G2 (Table 3).

Spearman’s correlation and regression (R^2^) analyses of G1 showed a positive correlation between HGMB-1 and R-OOHs (*p* = 0.04; r = 0.35). A negative correlation was detected between HGMB-1 and SHp (*p* < 0.001; r = −0.45), and between HGMB-1 and OXY (*p* = 0.02; r = −0.28) (Table 4).

In addition, negative correlations were found between SHP and the number of red blood cells (*p* = 0.037; r = 0.21) and hemoglobin (*p* = 0.042; r = 0.38) (Table 5). No significant correlation was detected between the oxidative stress variables and the clinical scores (Table 5).

## 4. Discussion

Intestinal parasites are considered a severe health problem in dogs, especially in puppies [35,36], which may display retarded growth, immunosuppression, and susceptibility to infectious diseases and generalized clinical illness [37].

In addition, intestinal parasites remain a significant threat to animal health in shelter environments, where the prevalence of gastrointestinal parasites is very high and often correlated with the onset of other parasitic and infectious diseases [13,38].

Oxidative stress is involved in the pathophysiology of numerous life-threatening diseases, and it is considered a potential pathogenic factor in the display of infectious and parasitic diseases in dogs [26,27,39].

Here, the relationship between gastrointestinal nematodes and the oxidative status in naturally infected dogs was investigated for the first time in order to define the role of oxidative stress in the onset of disease.

Oxidative stress occurs as a result of an imbalance between the accumulation of free radicals of oxygen-reactive species produced during aerobic metabolism, inflammation, and infections and the ability of tissues to detoxify the reactive products [17].

R-OOHs play a key role in the onset of oxidative stress responses in cells. They are generated as by-products of oxygen metabolism, and are capable of causing damage to essential biomolecules present in cells, such as DNA, proteins, and lipids, which compromises normal cellular functions [39]. Oxidation products can be transported systemically throughout the host’s body and reflect the local and systemic status of oxidative stress.

R-OOHs perform microbicidal activities as part of the host’s defense mechanism against parasitic pathogens, as they are capable of increasing the susceptibility of pathogens to phagocytic killing within the host’s tissues [40]. The production of ROOHs is one of the first lines of defense against the presence of parasites. When parasites infect the host, the immune system responds with a massive production of ROOHs, oxidation triggered by macrophages, and phagocyte activation [40]. Different studies have reported a higher susceptibility to infections in subjects with defective antioxidative mechanisms [39].

In fact, during the invasion process, parasites interact with the free radicals of the host, which are produced by the activation of macrophages and neutrophils to combat pathogens [17]. The entire process is carried out in the host cells as an effect of the parasite, and is influenced by the interaction between the pathogen and redox-active antiparasitic molecules. To preserve redox homeostasis and eliminate ROS, enzymatic/nonenzymatic antioxidant processes are put in place by the organism to inhibit or interrupt the formation of pro-oxidants, as well as to repair and substitute injured macromolecules. Thiols are a group of organic sulfur derivatives that are essential for the correct function of the biological system, and are considered markers of balance in both intracellular and extracellular oxidative processes. They play a central role in controlling the redox state of the cell, influencing the antioxidative capacity of cells through the definition of protein structure, the regulation of enzyme activity, and the control of transcription activity [41]. Low SHp levels were observed in dogs infected by gastrointestinal nematodes (G1), suggesting an oxidative imbalance status, as has been speculated elsewhere [42]. A lack of thiols has been widely reported in viral, bacterial, and parasitic diseases [26,41,43], indicating that thiol oxidation decreases in response to the increasing levels of total oxidant status.

The results reported a significant level of HMGB-1 in the dogs with nematode infections, attributable to their central immune response. An increase in HMGB-1 has been reported in different nematode infections in humans [44] and sheep [45].

HMGB-1 is also released by parasites during the infection process, and it has been documented that it may play an important function in the treatment response [44].

HGMB-1 is a chromosomal protein involved in all DNA transaction actions [46,47]. In addition, HMGB-1 is important in immunological activities, such as the induction of cytokine production, cell proliferation, chemotaxis, and differentiation [48,49]. In fact, it is secreted as a multifunctional alarmin [50,51,52,53] from necrotic cells [18] following an apposite stimulus [54], and it activates the release of inflammatory mediators.

Dogs infected by gastrointestinal nematodes showed an inflammatory response that tended to be of the T-helper type 2 (Th2), indicating that this immune evasion promotes parasite survival, limiting the development of pathological lesions resulting from aggressive, proinflammatory responses. Helminthic invasions injure the host’s epithelial tissue and lead to the release of host-damage-associated molecular patterns [55,56], including S100 small calcium-binding proteins such as HMGB-1.

The clinical signs of intestinal parasites are often nonspecific and vague. Malabsorption, diarrhea, hemorrhage, and a reduced growth rate are the clinical signs most frequently detected during the course of helminthiasis, becoming more severe in the presence of other secondary diseases [57].

The dogs belonging to G1 showed significantly lower values of RBC, Hgb, and PCV, suggesting the presence of anemia. Anemia is considered a predominant laboratory finding during the course of helminthiasis [37]. Although its occurrence may be strongly related to the activity of hematophagous parasites that cause traumatic lesions to the host’s intestinal mucosa [58], it seems that oxidative stress intervenes in several mechanisms involved in its onset [59]. The negative correlation between RBC, Hgb concentration, and SHp could suggest that oxidative stress contributes to the onset of anemia in dogs affected by helminthiasis. Presumably, these findings are related to altered eryptosis that is not adequately compensated by erythropoiesis, which is observed in cases of oxidative stress [60,61].

In addition, an inadequate oxidative imbalance is a predictor of enhanced hemolysis as a consequence of erythrocyte membrane damage [62].

## 5. Conclusions

Parasites perform complicated life cycles in their host, during which they are exposed to oxidative processes induced by the host’s immune system and endogenous activities. The results reported here suggest that helminthiasis is associated with oxidative stress in dogs and an increase in HGMB-1. Comprehending the dynamics of reactive oxygen species in the regulation of cellular redox homeostasis is very important for considering further biomedical approaches to novel drug development.

## Figures and Tables

**Table 1 antioxidants-11-01679-t001:** The scoring system for the clinical assessment of included dogs.

Variables	0	1
Mucous membranes	Pink	Pale
Ill thrift	Absent	Present

Haircoat		
	Normal	Poor
Dehydration	<5%	<5%
Melena	Absent	Present
Diarrhea	Absent	Present
Weight loss	Absent	Present
Vomiting	Absent	Present
Abdominal swelling	Absent	Present
Hyperthermia	<39.7 °C	>39.7 °C
Lethargy	Absent	Present

**Table 2 antioxidants-11-01679-t002:** Hematological and biochemical variables in dogs with gastrointestinal nematodes (G1) and in healthy dogs (G2).

Variable	Unit	G1		G2		Reference Ranges [33,34]	*p*-Value
		Mean	SD	Mean	SD		
**RBC**	(10^6^/μL)	4.2	±1.6	6.7	±2.1	5.6–8.7	0.042
**Hgb**	(ng/mL)	11.9	±1.8	15.3	±1.9	14.7–17.7	0.038
**PCV**	(%)	35.1	±4.7	47	±4.2	42–53	0.016
**WBC**	(10^3^/μL)	16.2	±5.4	8.1	±2.6	4.6–10.6	0.027
**PLT**	(10^3^/μL)	227	±124	290	±99	150–400	Ns
**BUN**	(mg/dL)	14.6	±10.8	11.4	±8.3	5–21	Ns
**CREA**	(mg/dL)	1.5	±0.4	1.3	±0.3	0.3–1.2	Ns
**AST**	(U/L)	45.1	±17.4	27.6	±13.3	0–40	Ns
**ALT**	(U/L)	29.4	±8.4	34.4	±15.6	0–40	Ns
**ALB**	(g/dL)	3.1	±0.71	3.9	±0.56	3.0–4.4	0.045
**GLOB**	(g/dL)	3	±0.01	2.7	±0.48	1.8–3.9	Ns
**TP**	(g/dL)	5.9	±0.3	6.6	±0.4	6.4–7.9	0.046

RBC, red blood cells; Hgb, hemoglobin; PCV, packed-cell volume; WBC, white blood cells; PLT, platelets; BUN, blood urea nitrogen; CREA, creatinine; AST, aspartate-amino transferase; ALT, alanine-amino transferase; ALB, albumin; GLOB, globulin; TP, total protein.

**Table 3 antioxidants-11-01679-t003:** R-OOHs, OXY, SHp, and HGMB-1 concentrations in G1 and G2 groups.

Variable	Unit	G1		G2		*p*-Value
		Mean	SD	Mean	SD	
**R-OOHs**	(CARR U)	259.58	±73.78	198.55	±40.61	0.014
**OXY**	(μmol HCLO/mL)	187.63	±161.67	65.59	±16.34	Ns
**SHp**	(μmol/L)	147.80	±74.16	239.33	±107.88	0.047
**HGMB-1**		12.6	±10.37	5.9	±2.9	0.046

G1, dogs with a diagnosis of gastrointestinal nematodes; G2, healthy dogs; Ns, statistically non-significant differences.

**Table 4 antioxidants-11-01679-t004:** The correlation between considered variables of oxidative stress in the G1 group (Spearman’s rank order) (* *p*< 0.05; ** *p* < 0.01).

	R-OOHs	OXY	SHp	HGMB-1
**R-OOHs**	1.0	−0.18	−0.46	0.35
**OXY**	−0.18	1.0	−0.71	−0.28 *
**SHp**	−0.46	−0.71	1.0	−0.45 **
**HGMB-1**	0.06 *	−0.28 *	−0.45 **	1.0

**Table 5 antioxidants-11-01679-t005:** Correlation between oxidative stress, hematochemical variables, and the clinical score in the G1 group (Spearman’s rank order) (* *p* < 0.05).

	RBC	Hgb	PCV	WBC	PLT	ALT	AST	BUN	CREA	ALB	GLOB	TP	CS
**R-OOHs**	0.21	0.43	0.78	0.83	0.90	0.03	0.01	0.42	0.23	−0.19	0.02	−0.21	−0.16
**OXY**	−0.21	−0.05	−0.01	0.54	0.31	−0.10	0.29	0.09	0.21	−0.09	0.23	−0.41	−0.01
**SHp**	−0.21 *	−0.38 *	−0.32	−0.28	0.05	0.04	0.02	0.10	−0.11	0.02	−0.09	0.38	−0.09
**HGMB-1**	0.02	0.02	0.11	−0.13	−0.63	0.25	−0.10	0.12	0.08	0.31	0.34	0.11	−0.08

R-OOHs, reactive oxidative metabolites; OXY, antioxidant barrier; SHp, thiol groups of plasma compounds; RBC, red blood cells; Hgb, hemoglobin; PCV, packed-cell volume; WBC, white blood cells; PLT, platelets; ALT, alanine-amino transferase; AST, aspartate-amino transferase; BUN, blood urea nitrogen; CREA, creatinine; ALB, albumin; GLOB, globulin; TP, total protein; CS, clinical score.

## Data Availability

Data is contained within the article.

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
