# Peer review of "Oxidative Stress and High-Mobility Group Box 1 Assay in Dogs with Gastrointestinal Parasites"

_antioxidants, 2022, doi:10.3390/antiox11091679_

Round 1
Reviewer 1 Report
The manuscript is interesting and well written, being one of the first observation on the involvement of oxidative stress and HGMB1 during parasites infection, it is worth of publication. Nevertheless, some important concerns must be addressed before the manuscript can be accepted.
The phone number of the corresponding author seems incomplete, please check.
English language is fine, but an accurate revision is required. Some corrections are depicted below.
It is critical that the authors allow the reader to repeat the experiments by giving all needed information to this purpose.
L2 – The authors stated (L 26-27) that these are preliminary results, this is right mainly because no differences between the various parasite infections was investigated, but the title should be changed accordingly. I suggest adding “preliminary results” in the title.
L76 – replace “is” with “was”
L89 – delete “included”
L95 – add “a”, after Italy
L102 – replace “were” with “was”
L110 – Table 1, in the line referred to Dehydration variable replace 1 “< 5%” with “> 5%” (third column, under score 1)
L115 – replace “stored” with “divided”
L119 – The authors described the generic assay of R-OOHs but, at line 123, they write “dROMs were assayed….”. Thus, I assume that the reagents from Diacron were used for the evaluation of oxidative status. In any event, more information concerning the reagents, or the procedures must be added in the text. Useful information can be found in: Chiofalo, B. et al., Effects of dietary protein and fat concentrations on hormonal and oxidative blood stress biomarkers in guide dogs during training Journal of Veterinary Behavior, 2020, 37, pp. 86–92
L144, 149 – Similarly, the system and reagents used for CBC and clinical chemistry evaluation should be described.
L166 – replace “were” with “was” and put it after vulpis.
L169 – Delete “While”
L225-227 – The sentence is not clear, please re-write it
L253 – “a” central role
L264 – release should be “released”
L279 – replace “severe” with “severity”
L282 – replace “as” with “a”
Reviewer 2 Report
OVERALL COMMENTS
This study aimed to evaluate oxidative stress due to helminth infections in dogs naturally infected with several species of gastrointestinal parasites. Together with clinical examination and blood analysis, the Authors evaluated the concentration of reactive oxidative metabolites, antioxidant barrier, thiol groups of plasma compounds and the High Group Monility Box-1 in two groups of kennel dogs, positive and negative for the presence of intestinal parasites, respectively. The study provides some good insights to fill in the knowledge of parasitic infections and their implications on the host's health status. However, the English language must be reviewed by a native speaker. Therefore, in my opinion, the manuscript is suitable for publication in Antioxidants, pending very minor revisions.
MINOR COMMENTS
Abstract
Line 19: Please remove “the” before “control”.
Introduction
Lines 33-34: Modify the sentence accordingly: “…to parasitic infections, particularly those caused by gastro-intestinal helminths [1-9], with some…”
Line 37: “which” instead of “wich”
Line 38: Delete “reported that” and modify the sentence as follows “ …in which more than a third of the canine population was shedding eggs and cysts of endoparasites”
Line 41: “in particular” instead of “a particular”
Line 42: “may be threatened” instead of “should be threatened”
Line 46: Replace “in general” with “ Among the gastro-intestinal parasite species, whipworms…”
Line 46: Should be “are” instead of “were”.
Line 74: Modify the sentence accordingly “to the best of the Authors’ knowledge, no studies have evaluated…”
M&M
I believe it is important to specify whether or not the animals have been treated against gastrointestinal parasites (i.e. molecule used, frequency and last treatment). Considering that the idea of kennels and how animals are treated within these facilities may be different in other parts of the world, it is best to specify and avoid any kind of bias in the reader. Also, were the dogs treated at the end of the study?
Line 96: Delete “please”
Line 103: Delete “healthy” as the absence of gastrointestinal parasitic infections is not synonymous with good health (e.g., other undiagnosed infections or any other pathology).
Results
It would be interesting to have information on the infection prevalance of each species and whether the authors found a significant correlation between species or parasitemia load and oxidative stress level.
Table 1
The output of “dehydration” cannot be the same for 0 and 1.
